# SoftFlow: Probabilistic Framework for Normalizing Flow on Manifolds

**Hyeongju Kim, Hyeonseung Lee, Woo Hyun Kang, Joun Yeop Lee, Nam Soo Kim**
Department of Electrical and Computer Engineering and INMC,
Seoul National University,
Seoul, South Korea
{hjkim, hslee, whkang, jylee}@hi.snu.ac.kr, nkim@snu.ac.kr

## Abstract

Flow-based generative models are composed of invertible transformations between two random variables of the same dimension. Therefore, flow-based models cannot be adequately trained if the dimension of the data distribution does not match that of the underlying target distribution. In this paper, we propose SoftFlow, a probabilistic framework for training normalizing flows on manifolds. To sidestep the dimension mismatch problem, SoftFlow estimates a conditional distribution of the perturbed input data instead of learning the data distribution directly. We experimentally show that SoftFlow can capture the innate structure of the manifold data and generate high-quality samples unlike the conventional flow-based models. Furthermore, we apply the proposed framework to 3D point clouds to alleviate the difficulty of forming thin structures for flow-based models. The proposed model for 3D point clouds, namely SoftPointFlow, can estimate the distribution of various shapes more accurately and achieves state-of-the-art performance in point cloud generation.

## 1   Introduction

Ever since Dinh et al. (2014) first introduced Non-linear Independent Component Estimation (NICE) that exploits a change of variables for density estimation, flow-based generative models have been widely studied and have shown competitive performance in many applications such as image generation (Kingma & Dhariwal, 2018), speech synthesis (Prenger et al., 2019; Kim et al., 2018), video prediction (Kumar et al., 2019) and machine translation (Ma et al., 2019). With this success, flow-based models are considered a potent technique for unsupervised learning due to their attractive merits: (i) exact log-likelihood evaluation, (ii) efficient synthetic data generation, and (iii) well-structured latent variable space. These properties enable the flow-based model to learn complex dependencies within high-dimensional data, generate a number of synthetic samples in real-time, and learn a semantically meaningful latent space which can be used for downstream tasks or interpolation between data points.

There also have been some theoretical developments as well as various application of flow-based models in recent years. For example, unlike the conventional flow-based models which typically perform dequantization by adding uniform noise to discrete data points (e.g., image) as a pre-process for the change of variable formula (Dinh et al., 2016; Papamakarios et al., 2017), Flow++ (Ho et al., 2019) proposed to leverage a variational dequantization technique to provide more natural and smoother density approximator of discrete data. Another example is a continuous normalizing flow (CNF) (Chen et al., 2018; Grathwohl et al., 2018). While discrete flow-based models adopt a restricted architecture for ease of computing the determinant of the Jacobian, CNFs impose no

limits on the choice of model architectures since the objective function of CNFs can be efficiently calculated via Hutchinson's estimator (Hutchinson, 1990).

In this paper, we further aim to overcome another limitation of existing flow-based models, i.e., normalizing flows on manifolds. There have been some interesting works on the similar topic (Gemici et al., 2016; Rezende et al., 2020; Brehmer & Cranmer, 2020) but they may not guarantee numerical stability, or require the information about manifolds (e.g., structure and dimensionality) in advance or additional training steps to estimate it. Here, we propose a novel probabilistic framework for training normalizing flows on manifolds without any assumption and prescribed information. To begin with, we show that conventional normalizing flows cannot accurately estimate the data distribution if the data resides on a low dimensional manifold. To circumvent this issue, the proposed method, namely SoftFlow, perturbs the data with random noise sampled from different distributions and estimates the conditional distribution of the perturbed data. Unlike the conventional normalizing flows, SoftFlow is able to capture the distribution of the manifold data and synthesize high-quality samples. Furthermore, we also propose SoftPointFlow for 3D point cloud generation which relieves the difficulty of forming thin structures. We experimentally demonstrate that SoftPointFlow achieves cutting-edge performance among many point cloud generation models. Our framework is intuitive, simple and easily applicable to the existing flow-based models including both discrete and continuous normalizing flows. To encourage reproducibility, we attach the code for both SoftFlow and SoftPointFlow used in the experiments[1].

## 2 Flow-based generative model

A normalizing flow (Rezende & Mohamed, 2015) consists of invertible mappings from a simple latent distribution $p_Z(z)$ (e.g., isotropic Gaussian) to a complex data distribution $p_X(x)$. Let $f_i$ be an invertible transformation from $z^{i-1}$ to $z^i$, $z^0 = z$ and $z^n = x$ ($z^i \in \mathbb{R}^D$ for $i = 0, ..., n$). Then, the log-likelihood $\log p_X(x)$ can be expressed in terms of the latent variable $z$ following the change of variables theorem:

$$z = f_1^{-1} \circ f_2^{-1} \circ ... \circ f_n^{-1}(x), \tag{1}$$

$$\log p_X(x) = \log p_Z(z) - \sum_{i=1}^{n} \log \left| \det \left( \frac{\partial f_i}{\partial z^{i-1}} \right) \right|. \tag{2}$$

Eqs. (1) and (2) suggest that the optimization of flow-based models requires the tractability of computing $f_i^{-1}$ and $\log \left| \det \left( \frac{\partial f_i}{\partial z^{i-1}} \right) \right|$. After training, sampling process can be performed efficiently as follows:

$$z \sim p_Z(z), \tag{3}$$

$$x = f_n \circ f_{n-1} \circ ... \circ f_1(z). \tag{4}$$

Recently, Chen et al. (2018) introduced a continuous normalizing flow (CNF) where the latent variable is assumed to be time-varying and the change of its log-density follows the instantaneous change of variables formula. More specifically, continuous-time analogs of Eqs. (1) and (2) can be given by

$$z(t_0) = z(t_1) + \int_{t_1}^{t_0} f(z(t), t)dt, \tag{5}$$

$$\log p(z(t_1)) = \log p(z(t_0)) - \int_{t_0}^{t_1} \text{Tr} \left( \frac{\partial f(z(t), t)}{\partial z(t)} \right) dt, \tag{6}$$

where $f(z(t), t) = \frac{dz(t)}{dt}$, $z(t_0) = z$ and $z(t_1) = x$. Unlike conventional normalizing flows, CNFs impose no restriction on the choice of model architecture since the trace operation in Eq. (6) can be efficiently computed using the Hutchinson's estimator (Grathwohl et al., 2018) and the sampling process is performed by reversing the time interval in Eq. (5). However, due to the large computational cost of the ODE solver, CNFs usually require a long time for training (e.g., Grathwohl et al. (2018) reported that they trained the CNF on MNIST for 5 days using 6 GPUs).

# 3 Normalizing flow on manifolds

Although normalizing flows have shown promising results on various tasks such as image generation (Kingma & Dhariwal, 2018), voice conversion (Serrà et al., 2019) and machine translation (Ma et al., 2019), current flow-based models are not suitable for estimating the distribution of the data sitting on a lower-dimensional manifold. We note that Eqs. (2) and (6) are valid only when the data distribution and the target distribution have the same dimensions. To demonstrate this limitation, we trained 2 FFJORD models (Grathwohl et al., 2018) on different data distributions and present the results in Fig. 1 where the left column represents the data distribution, the central column represents the scatter plot of the corresponding latent variables warped from the data points through the trained networks, and the right column denotes the target latent distribution that we initially set for training. When the dimension of the data distribution matches to that of the target distribution, the FFJORD model properly transforms the data points into the latent points. However, when the FFJORD model is trained on 1D manifold data scattered over 2D space, the distribution of the latent variables corresponding to the data points is quite different from the target latent distribution. This simple experiment exhibits the shortcoming of the current normalizing flows that they cannot expand the 1D manifold data points to the 2D shape of the target distribution since the transformations used in flow networks are homeomorphisms (Dupont et al., 2019). If the transformed latent variables cannot represent the whole 2D space, it is unclear which part of the data space would match the latent points outside the lines. The observation suggests that training the normalizing flows on manifolds according to Eq. (2) or Eq. (6) may result in degenerated performance.

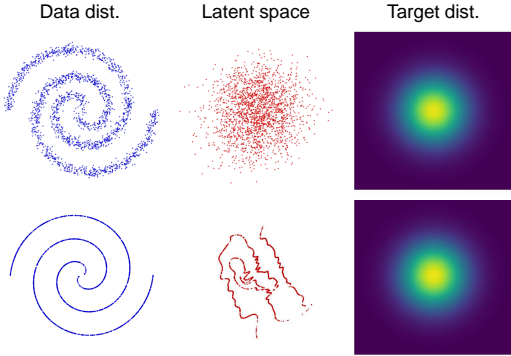

Figure 1: Illustration of normalizing flow trained on 2D data distribution (top) and 1D manifold data distribution (bottom).

In fact, the change of variables theorem used in Eq. (2) is not useful anymore if the dimension of the domain is lower than the dimension of the image. Let $F$ be a function from the contented subset $A \subset \mathbb{R}^m$ to a manifold $M \subset \mathbb{R}^n$ where $m < n$ as shown in Fig. 2. If $z \in A$ and $x \in M$ satisfy $F(z) = x$, the $m$-dimensional infinitesimal volume $dV_x$ around $x$ is given by

$$dV_x = \sqrt{\det(DF)^\dagger DF} dV_z \qquad (7)$$

where $DF$ is the $n \times m$ Jacobian matrix (i.e., $DF_{ij} = \frac{\partial x_i}{\partial z_j}$), $dV_z$ is the infinitesimal volume

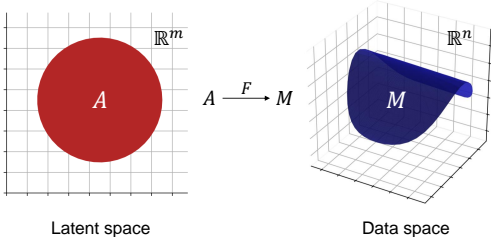

Figure 2: Example of the function that maps the contented subset of $\mathbb{R}^m$ to the manifold of $\mathbb{R}^n$.

around $z$, and $\dagger$ represents the transpose operation (Gemici et al., 2016; Ben-Israel, 1999). Therefore, the log-likelihood $\log p_X(x)$ can be computed as follows:

$$\log p_X(x) = \log p_Z(z) - \frac{1}{2} \log(\det(DF)^\dagger DF). \qquad (8)$$

Unfortunately, however, it is not straightforward to design a flow-based model according to Eq. (8) for a few reasons. First, transforming $x$ to $z$ is problematic as $F$ cannot be invertible in general. This prevents the use of maximum likelihood since flow-based models cannot be optimized according Eq. (8) without knowing $z$. Secondly, we are no longer able to employ the trick of setting the Jacobian to a lower triangular matrix as in the general flow models. It is because that $\det(DF)^\dagger DF$ is always a symmetric matrix. This restriction may lead to $\mathcal{O}(m^3)$ computational cost for the determinant. Finally, we need prior knowledge on the dimension of the manifold data in order to exactly determine $m$. Otherwise, we may rely on a heuristic search for $m$. These difficulties motivated us to come up with a novel probabilistic framework for training normalizing flows on manifolds which is presented in the following section.

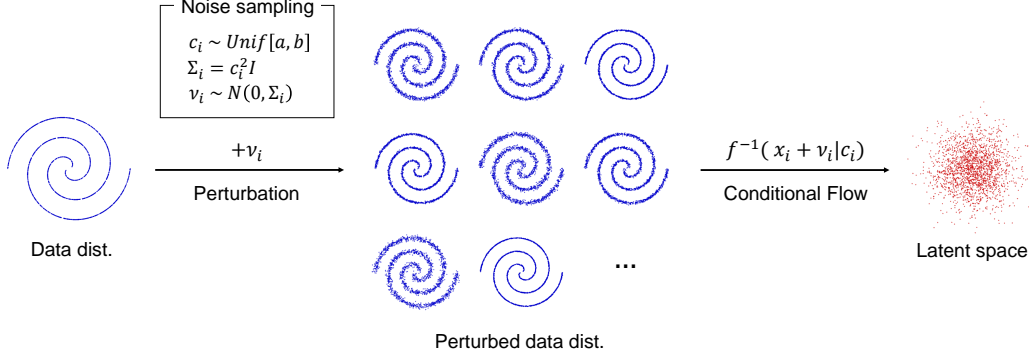

Figure 3: Proposed technique for training a normalizing flow on manifold data.

## 4  SoftFlow

Our ultimate goal is to appropriately train the normalizing flows on manifolds and generate realistic samples. The main cause of the aforementioned difficulties is the inherent nature of normalizing flows that the network output is homeomorphic to the input. To bridge the gap between the dimensions of the data and the target latent variable, we propose to estimate a conditional distribution of the perturbed data. The key idea is to add noise sampled from a randomly selected distribution and use the distribution parameters as a condition. For implementation, we perform the following steps for the $i$-th data point $x_i$: First, we sample a random value $c_i$ from the uniform distribution $unif[a, b]$, and set the noise distribution to $\mathcal{N}(0, \Sigma_i)$ where $\Sigma_i = c_i^2 I$. Next, we sample a noise vector $\nu_i$ from $\mathcal{N}(0, \Sigma_i)$ and obtain the perturbed data point $x_i'$ by adding $\nu_i$ to $x_i$. Note that now the distribution of $x_i'$ is not confined to a low dimensional manifold due to the addition of $\nu_i$. Let $f(\cdot|c_i)$ be the flow transformation from the latent variable $z$ to $x_i'$ (i.e., $f(z|c_i) = x_i'$), then the final objective function is given by

$$\log p_X(x_i'|c_i) = \log p_Z(z) - \log \left| \det \left( \frac{\partial f(z|c_i)}{\partial z} \right) \right|. \tag{9}$$

A summary of the proposed training procedure is illustrated in Fig.3. Since the support of the perturbed data distribution spans the entire dimensions of the data space, the normalizing flow on manifolds can be reliably trained according to Eq. (9). In addition, during training, the flow networks observe various distributions with different volumes and learn to transform the randomly perturbed data points into the latent variables properly. This enables the flow networks to understand and generalize the relation between the shape of data distributions and the noise distribution parameters. As a result, we can synthesize a realistic sample $x_{sp}$ by setting $c_{sp}$ to a small value or even zero as follows:

$$z_{sp} \sim p_Z(z), \tag{10}$$

$$x_{sp} = f(z_{sp}|c_{sp}). \tag{11}$$

Furthermore, it is obvious that the method can be extended to any CNF by adopting the following dynamics:

$$\frac{dz(t)}{dt} = f(z(t), t, c_i), \quad z(t_0) = z, \quad z(t_1) = x_i + \nu_i. \tag{12}$$

The proposed framework, namely *SoftFlow*, provides a new way to exploit a normalizing flow for manifold data. SoftFlow overcomes the dimension mismatch by estimating a perturbed data distribution which is conditioned on noise parameters. Both the training and sampling processes can be easily implemented within the existing flow-based frameworks.

### 4.1  Experiments on artificial data

We conducted a set of experiments in order to validate the proposed framework. To implement SoftFlow within the FFJORD architecture, we augmented another dimension for a noise distribution parameter to the conditioning networks. During training, we sampled $c_i$ from $Unif[0, 0.1]$ and

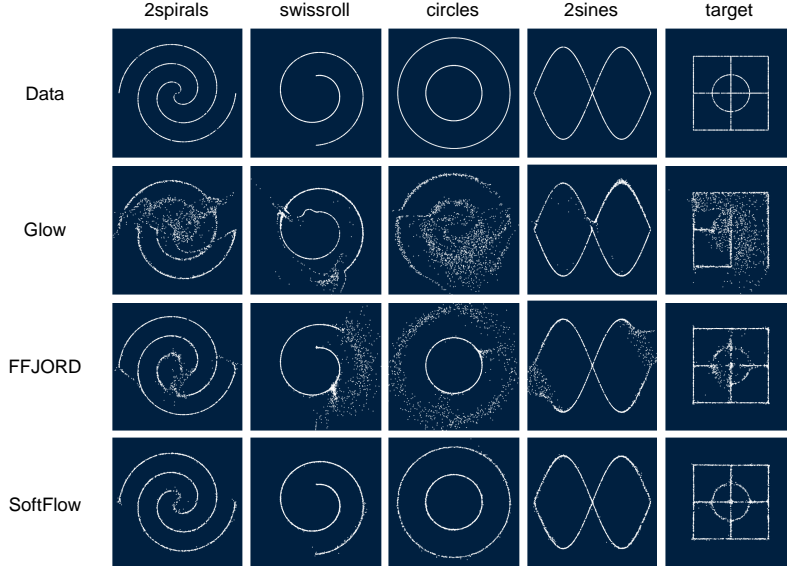

Figure 4: Samples from SoftFlow, Glow, and FFJORD trained on 5 different distributions.

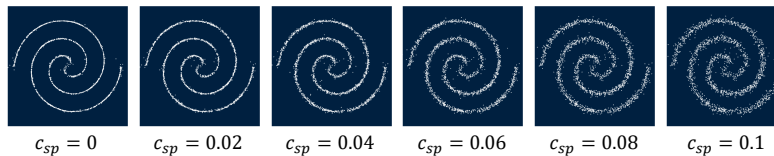

Figure 5: Examples of synthetic data points sampled from SoftFlow with different values of $c_{sp}$.

perturbed each data point $x_i$ by adding $\nu_i$ which was drawn from $\mathcal{N}(0, c_i^2 I)$. We scaled up $c_i$ by multiplying 20 to get $c_i^{in}$ and passed $c_i^{in}$ to the CNF networks for conditioning. We employed the same way of conditioning time $t$ in FFJORD for conditioning $c_i^{in}$. SoftFlow and FFJORD were trained on the data sampled from 5 different distributions[23] using the Adam optimizer (Kingma & Ba, 2014) with learning rate $10^{-3}$ for 36K iterations. We also trained a 100-layer Glow model for 100K iterations. All the distribution shapes were set to be composed of only 1D lines to exclude volume components.

The generative performance of SoftFlow, Glow, and FFJORD is shown in Fig. 4. For sampling, $c_{sp}$ in SoftFlow was set to zero. We can observe that Glow showed the worst performance in most distributions, and some parts of the sample clusters generated by FFJORD failed to fit the data distribution. Especially in the case of the circles distribution, both Glow and FFJORD generated poor samples which were scattered around the circles and formed a curved line connecting the inner and outer circles. The results demonstrate that Glow and FFJORD suffer from difficulties in estimating the distribution of the manifold data and cannot synthesize appropriate samples that agree with the data distribution. In contrast, SoftFlow is optimized according to the adequate objective function for manifold data. As a result, SoftFlow is capable of generating high-quality samples that follow the data distribution quite perfectly.

To examine how well the proposed model understands and generalizes the relation between the shape of a distribution and a noise distribution parameter, we drew different groups of samples obtained from SoftFlow by varying $c_{sp}$ from 0 to 0.1. As shown in Fig. 5, SoftFlow generated various samples which faithfully follow different distributions as we intended. The experimental results imply that SoftFlow can be further exploited to estimate an unseen distribution or produce a plausible synthetic distribution.

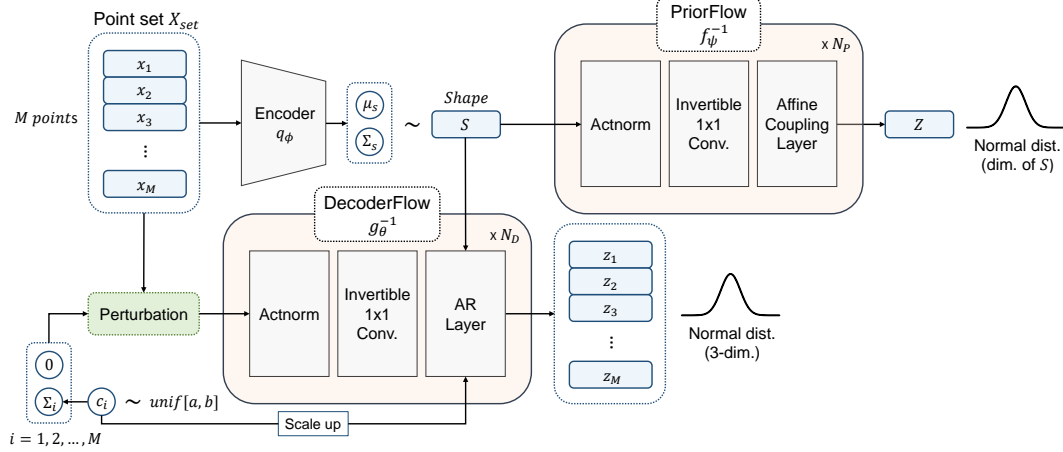

Figure 6: Schematic block diagram of SoftPointFlow.

# 5    SoftPointFlow

3D point clouds are a compact representation of geometric details which leverage sparsity of the data. Point clouds are becoming popular and attractive since they are processed by simple geometric operations and can be efficiently acquired by using various range-scanning devices such as LiDARs. In light of the benefits of point clouds, some recent works have proposed generative models for point clouds (Yang et al., 2019; Achlioptas et al., 2017; Groueix et al., 2018). However, generative modelling of point clouds is still a challenging task due to the high complexity of the space of point clouds. PointFlow (Yang et al., 2019) is the current state-of-the-art model for point cloud generation but still had difficulty forming thin structures. We assume that the difficulty stems from the inability of normalizing flows to estimate the density on lower-dimensional manifolds, and propose SoftPointFlow to mitigate the issue by applying the SoftFlow technique to PointFlow.

The overall architecture of SoftPointFlow[4] is shown in Fig. 6. SoftPointFlow models two-level hierarchical distributions of shape and points, following the same training framework of PointFlow. Given a point set $X_{set}$ consisting of $M$ points, we first encode $X_{set}$ into a latent variable $S$ using the reparameterization trick (Kingma & Welling, 2013). The encoder employs the same architecture as in PointFlow. We provide a more expressive and learnable prior for $S$ by employing $\mathrm{PriorFlow}$, a Glow-like architecture for estimating the likelihood of $S$. Each $x_i$ is randomly perturbed as follows:

$$x_i' = x_i + \nu_i, \quad \nu_i \sim \mathcal{N}(0, c_i^2 I), \quad c_i \sim unif[a, b]. \tag{13}$$

The perturbed point $x_i'$ goes through DecoderFlow to compute the conditional likelihood given $S$ and $c_i$. DecoderFlow adopts an autoregressive function for flow transformation (AR Layer) which offers parallel computation for training. Even though AR Layer requires serial operations for sampling, the sampling speed will not be degraded significantly since AR Layer only processes a 3-dimensional vector. Our final objective function $\mathcal{L}(X_{set}; \theta, \psi, \phi)$ is given as follows:

$$\mathcal{L}(X_{set}; \theta, \psi, \phi) = \mathbb{E}_{q_\phi(S|X_{set})}[\log p_\theta(X_{set}|S) + \log p_\psi(S) - \log q_\phi(S|X_{set})]$$

$$\approx \mathbb{E}_{q_\phi(S|X_{set})}\left[\sum_{i=1}^{M}\left(\mathbb{E}_{c_i \sim unif[a,b]}\left[\log p(z_i) - \log\left|\det\left(\frac{\partial g_\theta(z_i|S, c_i)}{\partial z_i}\right)\right|\right]\right)\right.$$

$$\left. + \log p(Z) - \log\left|\det\left(\frac{\partial f_\psi(Z)}{\partial Z}\right)\right|\right] + H[q_\phi(S|X_{set})],$$

(14)

where $Z = f_\psi^{-1}(S)$, $z_i = g_\theta^{-1}(x_i'|S, c_i)$, and $H$ represents the entropy.

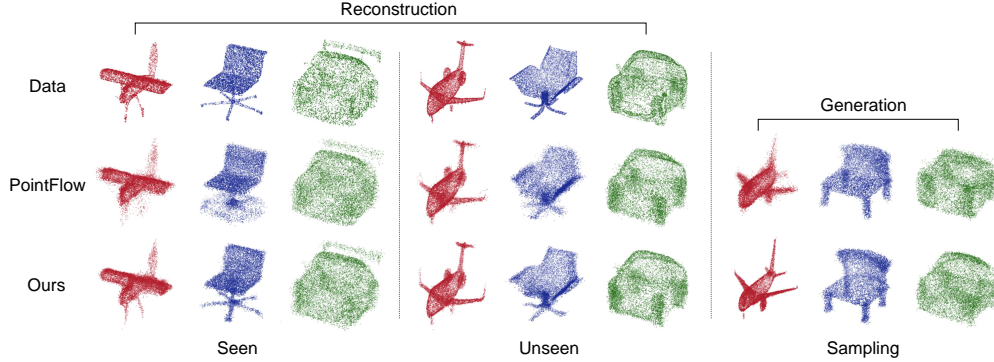

Figure 7: Examples of point clouds generated by PointFlow and SoftPointFlow. From left to right: reconstructed samples of seen data, reconstructed samples of unseen data, and synthetic samples.

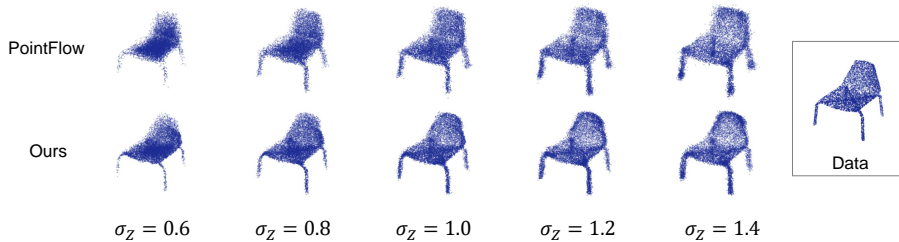

Figure 8: Reconstructed samples transformed from different latent distributions.

## 5.1 Experiments on point clouds

We conducted a set of experiments using the ShapeNet Core dataset (v2) (Chang et al., 2015) to evaluate the proposed framework for 3D point clouds. Three different categories were used for the experiments: airplane, chair, and car. We followed the same configuration for the training and test sets as in Yang et al. (2019), and used 5K points per shape in the training set to construct the validation set. We trained SoftPointFlow for 15K epochs with a batch size of 128 using four 2080-Ti GPUs. The initial learning rate of the Adam optimizer was set to $0.002$ and decayed by half after every 5K epochs. Each point set $X_{set}$ was compoesd of 2048 points ($M = 2048$).

SoftPointFlow was built on two discrete normalizing flow networks, $\mathrm{PriorFlow}$ and $\mathrm{DecoderFlow}$. $\mathrm{PriorFlow}$ consisted of 12 flow blocks of an actnorm, an invertible 1x1 convolution, and an affine coupling layer ($N_p = 12$). For the affine coupling layer, $\mathrm{PriorFlow}$ employed 4 convolution layers with gated `tanh` nonlinearities as well as residual connections and skip connections with 256 channels. The latent variable $S$ was squeezed to have 8 channels before going through $\mathrm{PriorFlow}$ and 2 of the channels were factored out after every 4 flow blocks following the multi-scale architecture (Dinh et al., 2016). On the other hand, $\mathrm{DecoderFlow}$ consisted of 9 flow blocks of an actnorm, an invertible 1x1 convolution, and an autoregressive (AR) layer ($N_D = 9$). Each AR layer employed 3 linear layers with gated `tanh` units. We set the number of channels in the linear and gate layers to 256. A concatenated vector of the input point and the noise parameter was passed to the linear layers. Also, the latent variable $S$ was used as a global condition by going through the gate layers. During training, $c_i$ was sampled from $unif[0, 0.075]$ and scaled up to have a maximum value 2 for the AR layers. For sampling, $c_{sp}$ was set to zero.

We report some samples generated by PointFlow and SoftPointFlow in Fig. 7. The axis ratio was adjusted to highlight the difference between the samples. We can observe that PointFlow generated blurry samples that failed to form thin structures such as chair legs or wing tips. In contrast, SoftPointFlow captured fine details of an object well and produced high-quality samples. We provide various samples generated by each model in Appendices B and C.

What would happen if we sample latent variables from $\mathcal{N}(0, \sigma_z^2 I)$ with different values of $\sigma_z$, and transform them into the data space? If the latent variable is sampled from the vicinity of high

density (i.e., a low value of $\sigma_z$), we can expect that the corresponding points in the data space would be focused on the main part of an object (e.g., a chair body). In the opposite case, the points may be gathered around the thin structure (e.g., chair legs). To investigate the representations that each model learned, we generated various point sets by varying $\sigma_z$, and report the results in Fig. 8. The overall tendency is similar to what we expected. However, as $\sigma_z$ increases, we observe that PointFlow failed to capture an X-shaped structure beneath the chair body while SoftPointFlow produced points that form the X shape. Also, as $\sigma_z$ decreases, the samples generated by PointFlow are concentrated on the center of the chair while the point clouds of SoftPointFlow well preserve the whole structure. The results demonstrate that SoftPointFlow learns more desirable features from manifold data and is robust to the variance of latent variables.

In order to compare SoftPointFlow with other generative models, we measured 1-nearest neighbor accuracy (1-NNA) of SoftPointFlow. The 1-NNA represents the leave-one-out accuracy of the 1-NN classifier and its ideal score is 50% (Lopez-Paz & Oquab, 2016). To compute the 1-NNA, two different distance metrics, Chamfer distance (CD) and earth mover's distance (EMD), can be employed to measure the similarity between point clouds. The generation results on 1-NNA are shown in Table 1. The results of l-GAN (Achlioptas et al., 2017), PC-GAN (Li et al., 2018) and PointFlow are taken from Yang et al. (2019). In all categories, SoftPointFlow achieved the significantly better results than the GAN-based models. Compared to PointFlow, SoftPointFlow showed the competitive performance on the airplane and chair data sets, and recorded the slightly lower scores

Table 1: Generation results on 1-NNA (%). Lower is better.

| Category | Model | CD | EMD |
|---|---|---|---|
| Airplane | l-GAN (EMD) | 87.65 | 85.68 |
| | PC-GAN | 94.35 | 92.32 |
| | PointFlow | 75.68 | 75.06 |
| | SoftPointFlow | **70.92** | **69.44** |
| Chair | l-GAN (EMD) | 64.73 | 65.56 |
| | PC-GAN | 76.03 | 78.37 |
| | PointFlow | 60.88 | **59.89** |
| | SoftPointFlow | **59.95** | 63.51 |
| Car | l-GAN (EMD) | 69.74 | 68.32 |
| | PC-GAN | 92.19 | 90.87 |
| | PointFlow | **60.65** | **62.36** |
| | SoftPointFlow | 62.63 | 64.71 |

on the car data set. Since the proportion of thin structures in the car data set is relatively low, we believe the results still support the validity of the proposed framework.

We also conducted a preference test to evaluate the perceptual quality of samples. We randomly selected 60 point clouds for each category (airplane, chair, and car) and obtained the reconstructed point clouds from PointFlow and SoftPointFlow. We asked 31 participants to assess which sample is more similar to the reference and better in quality. Each question presented a reference point cloud and two reconstructed point clouds as shown in the left and center columns of Fig. 7 but the order was random. The results are shown in Fig. 9. Surprisingly, SoftPointFlow received better scores than PointFlow in all cases. In particular, Soft-

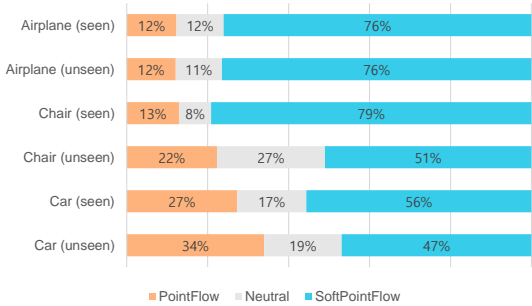

Figure 9: Results on preference test.

PointFlow outperformed by a large margin in the airplane shapes and the seen chair shapes. The overall results demonstrate that the proposed framework is considerably useful and suitable for modelling generative flows on point clouds.

## 6 Conclusion

In this paper, we have introduced a novel probabilistic framework, SoftFlow, for training a normalizing flow on manifolds. We experimentally demonstrated that SoftFlow is capable of capturing the innate structure of the manifold data and produces high-quality samples while the current flow-based models cannot. Also, we have successfully applied the proposed framework to point clouds to overcome the difficulty of modelling thin structures. Our generative model, SoftPointFlow, produced point clouds that describe more exactly the details of an object and achieved state-of-the-art performance for point

cloud generation. We believe that our framework can be further improved by theoretically identifying which noise distribution is more useful for training or by searching an architecture to leverage noise parameters efficiently.

## Broader Impact

This paper introduces a new way of designing a generative model for manifold data. This paper will motivate other researchers and engineers to employ the proposed framework for various applications. Like other generative models, the proposed model could produce biased samples if the training set is not properly set up. However, we believe that this paper will not cause a bad influence to the society in general use.

## Acknowledgments and Disclosure of Funding

This work was supported by Samsung Research Funding Center of Samsung Electronics under Project Number SRFC-IT1701-04.

## Footnotes

[1]https://github.com/ANLGBOY/SoftFlow

[2]The implementation of the data distributions can be found in our code.

[3]We provide the code for the 2sines and target distributions in Appendix A.

[4]The CNF networks in PointFlow are replaced with discrete normalizing flows for two reasons: (1) slow convergence of the CNF networks, and (2) validation of the proposed framework on discrete normalizing flows.

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
