[Supplementary Material 1]

# Appendices

## A    Code for the 2-sines and target distributions

```python
import numpy as np

def get_data_batch(bsz, dist):
    # bsz: batch size

    rng = np.random.RandomState()

    if dist == "2-sines":
        x = (rng.rand(bsz) - 0.5) * 2 * np.pi
        u = (rng.binomial(1, 0.5, bsz) - 0.5) * 2
        y = u * np.sin(x) * 2.5

    elif dist == "target":
        shapes = np.random.randint(7, size=bsz)
        mask = []
        for i in range(7):
            mask.append((shapes==i) * 1.) # boolean to float

        theta = np.linspace(0, 2 * np.pi, bsz, endpoint=False)

        x = (mask[0] + mask[1] + mask[2]) * (rng.rand(bsz) - 0.5) * 4 +\
         (-mask[3] + 0 * mask[4] + mask[5]) * 2 * np.ones(bsz) +\
         mask[6] * np.cos(theta)

        y = (-mask[0] + 0 * mask[1] + mask[2]) * 2 * np.ones(bsz) +\
         (mask[3] + mask[4] + mask[5]) * (rng.rand(bsz) -0.5) * 4 +\
         mask[6] * np.sin(theta)

    return np.stack((x, y), 1)
```

# B   Examples of generated point clouds

## B.1   Reconstruction of seen data

Figure 1: Reconstructed point clouds of seen data.

## B.2 Reconstruction of unseen data

Figure 2: Reconstructed point clouds of unseen data.

Figure 3: Synthetic point clouds generated by SoftPointFlow.

# C Point clouds generated by PointFlow and SoftPointFlow (original ratio)

## C.1 Reconstruction of seen data

Figure 4: Reconstructed point clouds of seen data.

## C.2 Reconstruction of unseen data

Figure 5: Reconstructed point clouds of unseen data.

## C.3  Generation

Figure 6: Synthetic point clouds generated by SoftPointFlow.



[Supplementary Material 2 · training_technique.pdf]



Noise sampling

$$c_i \sim Unif[a,b]$$
$$\Sigma_i = c_i^2 I$$
$$\nu_i \sim N(0, \Sigma_i)$$

Data dist.

$+\nu_i$

Perturbation

$f^{-1}(x_i + \nu_i | c_i)$

Conditional Flow

Latent space

...

Perturbed data dist.