[Reviews · NeurIPS 2020]

Review 1

Summary and Contributions: This work proposes a method for training normalizing flows on manifolds. The main issue with flow networks on manifolds is that the transformations are homeomorphisms. This work proposes to deal with this issue by adding noise to the data distribution such that it spans the input space. Evaluation is performed on 3D point cloud data.

Strengths: The main strengths are, 1. The proposed solution is simple, efficient and makes intuitive sense. 2. The examples on simpler 2D distributions illustrate the efficacy of the proposed solution. 3. The work includes helpful qualitative examples eg. Fig 7 and 8 esp on the airplane class. 4. The proposed method shows quantitative improvement on the airplane class which includes thin structures. 5. The paper is well written and easy to follow.

Weaknesses: ------------------------------------------------------------------------------- After rebuttal ------------------------------------------------------------------------------- The rebuttal addresses some of my concerns. But, * I am concerned about the general applicability of the work as only conditional log-likelhoods p(x | c_i ) can be computed using Eq. 9 -- not p(x). Moreover, results on standard benchmarks eg. CIFAR10 are not provided. * Also, my concerns about novelty esp. wrt. to [1] are not fully addressed. Note that [1] also adds noise to the input distribution (in particular the latent posterior) to ensure that the input distribution spans the target base distribution (in the latent space) and thus controls the contraction of the base distribution. However, I am not opposed to acceptance if the final version includes a discussion regarding similarities/differences wrt to [1]. ------------------------------------------------------------------------------- The main weakness are, 1. Limited novelty -- the proposed solution is very similar to the one proposed in [1], where (constant variance) noise was added to the latent vectors in the cVAE formulation. Although the proposed approach adds noise to the input data point, it is unclear which approach is better -- additional noise in the input space or latent space? 2. It is unclear why the performance wrt to PointFlow decreases in case of the Car class. It is clear that the Car class contains fewer thin structures vs airplane -- but ideally comparable performance is desirable. The proposed method should not harm performance in the absence of thin structures. Maybe this can be solved by instead adding noise in the latent space? 3. Additional experiments on more datasets esp. image datasets would increase the impact of the paper. 4. Section 3 should be reorganised to better convey the crucial points. [1] Conditional Flow Variational Autoencoders for Structured Sequence Prediction, NeurIPS Workshop 2019.

Correctness: Yes.

Clarity: For the most part. Section 3 can be organised better.

Relation to Prior Work: No. Crucial related work is not discussed -- Conditional Flow Variational Autoencoders for Structured Sequence Prediction, NeurIPS Workshop 2019.

Reproducibility: Yes

Additional Feedback: Please discuss relation to prior work and potential limitations e.g. the results on the Car class.


Review 2

Summary and Contributions: Proposes to learn approximate densities on manifolds by perturbing the data and learning a conditional flow conditioned on the perturbation level. The final flow is the conditional flow where the perturbation level is zero (or near zero). SoftFlow is demonstrated on 2D toy datasets and 3D point clouds (i.e., points on a manifold in 3D).

Strengths: - Proposes a simple way to model flows on manifolds by adding Gaussian noise and conditioning on the noise parameter. - Demonstrates empirically better performance on 2D datasets and point cloud datasets. - Human evaluation of point cloud generation from 31 participants (though seems more like a quick sanity check rather than carefully controlled study).

Weaknesses: The author response to the first question below was a little odd. The question did not ask about dequantizing image datasets though that could be an important point to discuss. Are the authors suggesting that dequantizing the image inherently avoids the manifold problem? The author response did not address comparison to other recent manifold flows or details on how the participants were selected. These are still unaddressed weaknesses. ---------- - No experiments on high dimensional datasets. Unclear if this is useful for more general datasets beyond 2D and 3D manifolds. Could this be useful for more general density estimation particularly high dimensional data (e.g., MNIST/CIFAR-10 images)? - It would be great to include some discussion on how your work compares to very recent manifold flows, e.g., [1] and [2], even though these are likely concurrent work. - Given that very little information is given about the human evaluation including how the participants were selected or how the experiment was conducted, it is a little unclear what this means (though it is still interesting). It is possible that these results are biased. [1] Lou, Aaron, et al. "Neural Manifold Ordinary Differential Equations." arXiv preprint arXiv:2006.10254 (2020). [2] Brehmer, Johann, and Kyle Cranmer. "Flows for simultaneous manifold learning and density estimation." arXiv preprint arXiv:2003.13913 (2020).

Correctness: Generally, this paper seems to be correct. The human evaluation study may be biased depending on how the participants were selected but is nonetheless interesting. I would add a qualification to the experimental sections that give evidence for the following claims but also soften the following claims a little bit since they are interpretations/intuitions: Lines 139-142 "In addition, during training, the flow networks observe various distributions with different volumes and learn to transform the randomly perturbed data points into the latent variables properly. This enables the flow networks to understand and generalize the relation between the shape of data distributions and the noise distribution parameters."

Clarity: Overall the paper is very well-written and easy to read.

Relation to Prior Work: Comparison and contrast with other very recent work (see weaknesses above) on manifold flows would be a great addition. I believe this work is likely concurrent with this work but it would be nice to include at least a discussion of similarities and differences.

Reproducibility: Yes

Additional Feedback: The author response was adequate. However, sampling standard deviation from the uniform is still seems to be an odd choice to me. ----- Why use the uniform for sampling the standard deviation of the noise? Why not exponential, gamma or chi-squared? Why sample c from [0, 0.1] and then multiply by 20 to pass into the FFJORD network? Why not just sample from [0, 2]?


Review 3

Summary and Contributions: The authors pointed out that flow-based models have a problem of not being able to train properly when the dimension of the data manifold is lower than that of the space, and proposed SoftFlow that estimates the conditional distribution of the perturbed input data with random noise. The effectiveness of the proposed model was confirmed through qualitative and quantitative experiments in comparison with conventional models.

Strengths: - The problem to be addressed in this study are clear. In particular, it explains theoretically why learning is difficult in flow-based models when the dimensions of space and manifold are different. - The proposed model is simple and easy to understand. The idea of adding a perturbation to mitigate the problem of dimensional differences is convincing.

Weaknesses: My concerns were almost resolved by author's responses. However, I think their response to my concern about why the authors don't evaluate the performance by estimating the likelihood is weak. They only explain the difference in how to evaluate the likelihood of PointFlow and SoftPointFlow, not a direct answer to my concern (i.e., the authors do not specify which of these differences is the reason for not comparing them). It is true that SoftPointFlow is trained on the conditional log-likelihood, but I think its log-likelihood can be evaluated by setting c_{sp} = 0. I still have the above concern, but this is not directly related to the main contribution of this paper and my other concerns have been resolved, so I raise the score for this paper.

Correctness: I think that the claims and methods in this study are correct.

Clarity: As mentioned above, the issues are clearly pointed out and the proposed model is simple and easy to understand.

Relation to Prior Work: Although there is no chapter on related studies in this paper, the authors fully explain the related studies in the introduction and in each chapter.

Reproducibility: Yes

Additional Feedback:


Review 4

Summary and Contributions: This paper addresses the important problem of fitting an invertible mapping between manifolds embedded in vector spaces of different dimensionality. Since these mappings need to be bijective, it is usually not possible to fit these manifolds since some density will always need to reside outside of them. The authors propose a very elegant solution by simply adding Gaussian noise to the manifold. The mapping is then conditioned on the scale of that noise and can thereby denoise the input and find a latent representation that presents only the manifold. I would also like to applaud the authors for running an actual study on the perceived quality of their generated sample. This is good scientific practice and makes their argument all the more convincing.

Strengths: The paper discusses the theory of normalizing flows on embedded manifolds and clearly illustrate the problem. They then motivate their approach which provides an easy way around these limitations. The efficacy of the approach is shown in various experiments which include point cloud generation. Besides offering metric-based generation results, the authors also conducted a study where participants had to rank generation from their model vs the PointFlow model. Their approach came out ahead on all comparisons.

Weaknesses: The main weakness I can see is that the impact of the noise distribution is not discussed in any detail and the ranges for its scale are simply stated. It would be nice to get some more insight into the impact of the noise.

Correctness: The claims made in the paper are correct and very well demonstrated by the empirical findings. The authors compare their method against several state of the art normalizing flow and models and show it to work on par if not better.

Clarity: The paper is very intelligible and offers a very good introduction to the problem. The author first convince the reader that the problem is worth solving before turning to their solution. Great work!

Relation to Prior Work: Prior work is discussed at great length and indeed exploited in the construction of their flow model. It is also clear how the work distinguishes itself from others. Here again, it helps that empirical comparison is done against recent state-of-the-art methods.

Reproducibility: Yes

Additional Feedback: 1) It seems to me that you are proposing a general methodology rather than a specific model. As such, I would be interested to know if your proposal works just as well with GLOW or if the FFJORD architecture is indeed needed. This may extend to a general comparison between discrete and continuous flows. I do understand that in theory you can apply noising to both types of flows but I'm curious if there's an empirical difference.

[Author Response · NeurIPS 2020]

Reviewer #1

[Issue] Limited novelty – the proposed method is very similar to CF-VAE.

- CF-VAE sets the noise variance to a fixed value to regularize the maximum amount of contraction of the base density.
However, SoftFlow uses different noise variances for every training data and sets the noise variance to a very small
value or zero when sampling. Our novelty comes from a two step noise sampling procedure: (1) sample a noise variance
$c_i$, (2) sample a noise $\nu_i$ from $\mathcal{N}(0, c_i I)$. Both the purpose and method of SoftFlow are quite different from those of
CF-VAE.

[Issue] It is unclear why the performance wrt to PointFlow decreases in case of the Car class.

- We believe that the actual performance is not degraded even in the case of the Car class. First of all, we want to point out
that there's still an ongoing discussion of evaluation metrics for point cloud generation and the 1-NNA is not a perfect
metric. To evaluate the perceptual quality, we conducted the preference test and SoftPointFlow outperformed in the
ALL classes. We provide various samples of SoftPointFlow in the appendix to support the performance improvement.

Reviewer #2

[Issue] Could this be useful for more general density estimation (e.g., MNIST/CIFAR-10 images)?

- A conventional normalizing flow (e.g., Masked Autoregressive Flow, Flow++) dequantize image by adding noise.
This allows image data to have continuous values and volume components in data space. Instead, we focused on point
clouds. Unlike pixels and voxels, point clouds have innately continuous values and cannot apply the same method as
in image (the original point clouds cannot be retrieved directly from the noise added one). Leveraging the proposed
method for more general density estimation can be studied in future work.

[Issue] Why use the uniform for sampling the standard deviation of the noise? Why not exponential, gamma or
chi-squared?

- We agree and plan to study the effect of noise variance choice. We mentioned that point in the conclusion section, 'Our
framework can be further improved by theoretically identifying which noise distribution is more useful for training.'.

[Issue] Why sample c from [0, 0.1] and then multiply? Why not just sample from [0, 2]?

- This is just rescaling to use a noise variance as a good input to neural networks. If we sample from [0, 2], the effect of
perturbation will be very high and the perturbed data will become too blurry.

Reviewer #3

[Issue] (Sec. 4.1) Why is the number of training iterations different in Glow and CNF?

- We trained them until they converged. Since CNF is more expressive and has much lower number of parameters than
Glow, CNF converges with relatively lower iterations.

[Issue] (Sec. 4.1) It is not explained what kind of distribution $p_Z(z)$ is. Is this isotropic Gaussian?

- We apply the proposed method to FFJORD. FFJORD employs isotropic Gaussian for $p_Z(z)$.

[Issue] (Sec. 4.1) Why is $c_i$ multiplied by 20 when conditioning on the CNF network?

- Since the value of $c_i$ is too small, we rescaled it to be [0, 2] for neural network input.

[Issue] (Sec. 5.1) Why is the range of samples from the uniform distribution very narrower than that of Sec 4.1?

- The range is determined taking into account the data distribution. Since data points of the ShapeNet is more compactly
distributed than artificial data (Sec 4.1), we narrowed the uniform distribution interval.

[Issue] (Sec. 5.1) In this experiment, the authors did not evaluate the performance by estimating the likelihood. Why?

- While PointFlow estimates log-likelihood by solving ODEs, SoftPointFlow computes the explicit conditional log-
likelihood in a deterministic way (i.e., a closed form solution). We think the direct likelihood comparison is not
appropriate.

[Issue] (Sec. 5.1) I am not sure that point clouds in 3D space meet the challenge addressed in this paper.

- Point clouds are usually produced by 3D scanners and are scattered over the surface of an object. If not perturbed,
point clouds often contain line or plane components. We believe that learning the distribution of point clouds is a good
application of the proposed method.

[Meta-Review · NeurIPS 2020]

This paper proposes to layer flows conditioned on the noise level of artificial noise added to the data. The goal is to address issues coming from data living on sub-manifolds of the input space. OK submission overall.